# Age Differences in Pacing in Endurance Running: Comparison between Marathon and Half-Marathon Men and Women

**DOI:** 10.3390/medicina55080479

**Published:** 2019-08-14

**Authors:** Ivan Cuk, Pantelis Theodoros Nikolaidis, Srdjan Markovic, Beat Knechtle

**Affiliations:** 1Faculty of Physical Education and Sports Management, Singidunum University, 11000 Belgrade, Serbia; 2Exercise Physiology Laboratory, 18450 Nikaia, Greece; 3School of Health and Caring Sciences, University of West Attica, 11244 Athens, Greece; 4Institute of Primary Care, University of Zurich, 8006 Zürich, Switzerland; 5Medbase St. Gallen Am Vadianplatz, 9000 St. Gallen, Switzerland

**Keywords:** aerobic endurance, running, pacing strategy, aging, health

## Abstract

*Background and Objective*: The increased popularity of marathons and half-marathons has led to a significant increase in the number of master runners worldwide. Since the age-related decrease in performance is dependent on race duration, pacing in long distance running might also vary by race distance in both men and women. Therefore, the main aim of this study was to assess pacing differences between marathon and half-marathon runners with regard to the runners’ age group, and independently for men and women. *Materials and Methods*: In total, 17,465 participants in the Vienna City marathon in 2017 were considered for this study (marathon, *N* = 6081; half-marathon, *N* = 11,384). Pacing was expressed as two variables (i.e., pace range and end spurt). *Results*: All runners showed positive pacing strategies (i.e., a fast start with gradual decrease of speed). However, marathon runners showed greater variability in pacing than half-marathon runners. Furthermore, women showed no differences in pace variability in regard to the age group, whereas men younger than 30 years of age, as well as older men (over the age of 60), showed a greater variability in pace than other age groups. Finally, younger half-marathon men and women showed the fastest end spurt compared to older age groups and marathon runners. *Conclusions*: The presented findings could help sports and medicine practitioners to create age specific training plans and pacing strategies. This approach could help long distance runners to improve their physical fitness, achieve better race times, reduce the potential risk of musculoskeletal injuries and increase the overall pleasure of long distance running.

## 1. Introduction

Pacing in long distance sport events, such as cycling, running, cross-country skiing or triathlon races, can be defined as the efficient distribution of energetic reserves, power output, and speed through the entire race, without a significant slowdown [1]. For successful long distance pacing, it is necessary to select a suitable pacing strategy. This can be defined as a self-selected strategy that long distance athletes adopt from the beginning of a race [2]. Several pacing strategies have been previously observed by researchers [3,4]. These can be broadly categorized as: negative pacing (slow start with gradual increase of speed), positive pacing (fast start with gradual decrease of speed), even pacing (without significant speed changes) and variable pacing (with significant speed changes). Selecting an optimal pacing strategy can be a crucial aspect in the successful completion of a long distance event [5,6]. Additionally, the best pacing strategies can decrease the potential risk of musculoskeletal injuries [4], improve the performance of an athlete [7] as well as increase the overall pleasure of event participation for recreational runners [6]. 

The most popular long distance events in recent decades are running events, such as the marathon and half-marathon. Pacing in marathons is, so far, rather well documented in regards to the age, sex, performance, physiological, psychological and neurological aspects [2,8,9,10,11]. In general, even pacing and negative pacing strategies have proved to be the best strategies for maximizing performance in prolonged activities [12,13]; however, studies have shown that marathon runners seem to prefer positive pacing, regardless of age, sex or performance. Positive pacing strategies are the result of psychological factors (e.g., fast start due to competitiveness) [8] as well as physiological factors, such as neurological fatigue and muscle glycogen depletion [1]. Physiological factors particularly influence men’s pacing strategies in long distance running in comparison to women’s pacing strategies. Men are more likely to exhaust muscle glycogen, since they have weaker fat utilization systems than women [13,14]. Therefore, future studies on pacing in long distance running (with regard to age or performance differences) should assess pacing independently in men and women. 

Furthermore, substantial muscle fatigue, inflammation and fiber damage are more significant in marathons compared to half-marathons [15,16]. As a result, in recent years, the half-marathon has become the preferred long distance event for runners. The increasing popularity of half-marathon running can be observed in the United States [17], Europe [18] and worldwide [19]. Besides the previously mentioned medical benefits of half-marathon running in comparison to marathon running, half-marathons require less time to prepare for and less time to complete, thus its popularity [6]. Even so, pacing in long distance running is not as well documented as marathon pacing, particularly in recreational athletes of all ages. However, pacing in half-marathon events, has shown less variability than pacing in marathons, with regards to the participants’ ages [19]. Since only one event/race was analyzed, with a somewhat limited number of participants, these results cannot be generalized. 

The increased popularity of both marathon and half-marathon events has led to a significant increase in participants, particularly master athletes, in both the USA and Europe [17,18,20]. Long distance running could provide considerable health benefits for older runners, such as: risk reduction of cardiovascular diseases, cancer, diabetes, depression, and falls [21]. Therefore, insight into pacing strategies for age group runners could be an important scientific breakthrough for sports and medicine practitioners specializing in treating master runners as well as younger runners. The currently available studies regarding pacing in age group endurance runners prove to be somewhat inconsistent in their findings. There are some indications that older marathon runners are more likely to utilize even or negative pacing strategies compared to younger runners [9,13]. Other studies have shown rather similar pacing profiles in age group marathon runners [19,22]. Differences in pacing might also vary by race distance, since age-related decrease in performance is dependent on race duration [23]. Finally, only one study has investigated pacing between age groups of half-marathon runners [19]. Half-marathoners in all age groups had more even pacing in comparison to marathoners, whereas age did not play an important role in the pacing. However, a reduced number of participants, as well as lack of information on weather conditions (e.g., wind, humidity, temperature) limits the findings in this study.

Considering all previously mentioned differences between marathon and half-marathon running (e.g., performance, physiological, psychological and neurological aspects), further studies on pacing in master runners for these long distance events are needed. Therefore, the main aim of this study was to assess pacing differences between marathon and half-marathon events in regards to the runners’ age group, independently for men and women.

## 2. Materials and Methods

### 2.1. Participants and Race Details

This study was approved by the Institutional Review Board of Kanton St. Gallen, Switzerland (Approval number EKSG 01-06-2010), with a waiver of the requirement for informed consent of the participants as the study involved the analysis of publicly available data. The study was conducted in accordance with recognized ethical standards according to the Declaration of Helsinki adopted in 1964 and revised in 2013. 

For the purpose of this study, we included official results and split times from the 2017 Vienna City Marathon [24], i.e., the initial sample. Participants who did not finish any of the races, or did not have record of any of the split times were excluded from the initial sample. Moreover, participants who did not provide information on their age were also excluded from the initial sample. In total, 17,465 participants were considered for this study (marathon, *N* = 6081; half-marathon, *N* = 11,384).

Further information regarding the 2017 Vienna City Marathon is as follows:Both the marathon and the half-marathon were held on the same day, on an officially certified and fairly flat track (the elevation difference was only 50 m; ranging from 154 to 210 m). For comparison, the Berlin Marathon, considered to be “the fastest marathon” has an elevation difference of 21 m [19].During the race day, the weather was cloudy, with temperatures ranging from 7.8 °C at 9 am to 11.8 °C at 2 pm, without excess humidity or strong wind [24].No additional information on humidity grade or wind speed was available on the official race website.The half marathon race was entirely contained within the marathon race.

### 2.2. Data Analysis

In the first step of data analysis, we calculated the average speed for the entire race for each participant in both the half-marathon and the marathon. Additionally, we calculated the average running speed in five race sections, for both the marathon and the half-marathon [6,25]. The race sections were divided as follows:

Section 1 included the average running speed from start to the 10th km of the marathon race and from start to the 5th km of the half-marathon race. Both distances correspond to the first 23.7% of the marathon and half-marathon races.

Section 2 included the average running speed from the 10th km to the 20th km of the marathon race and from the 5th km to the 10th km of the half-marathon race. These distances represent a section of 23.7–47.4% of the marathon and half-marathon races.

Section 3 included the average running speed from the 20th km to the 30th km of the marathon race and from the 10th km to the 15th km of the half-marathon race. Both distances represent a section of 47.4–71.1% of the marathon and half-marathon races.

Section 4 included the average running speed from the 30th km to the 40th km of the marathon race and from the 15th km to the 20th km of the half-marathon race. These distances represent a section of 71.1–94.8 % of the marathon and half-marathon races.

Section 5 (i.e., the end spurt), included the average running speed from the 40th kilometer, to the race finish (42.195 km) in the marathon as well as from the 20th kilometer to the race finish (21.0975 km) in the half-marathon. The end spurt represents a section of94.8% to the finish line of the marathon and half-marathon races.

Furthermore, considering the methodology of Breen et al. [26], we calculated the variable of interest, called the “pace range”. Five race sections were subsequently expressed as a percentage faster or slower than the average section speed. The fastest section for each individual was then named the “positive range” (PR), while the slowest segment was named the “negative range” (NR). In addition, the absolute sum of the positive range and negative range was calculated and named the “pace range” (PaceR). This method allowed for normalized speed comparisons between all athletes as well as between the marathon and half-marathon. 

Finally, to examine the final 2.195 km of the marathon and the final 1.0975 km of the half-marathon, running speed for this segment was expressed as a percentage faster or slower than the running speed during Section 4 (i.e., 30–40 km for the marathon and 15–20 km for the half-marathon). This variable aimed to examine the “end spurt” (ES) and was named as such.

### 2.3. Statistical Analysis

Prior to all statistical tests, descriptive statistics were calculated as the mean and standard deviation. Moreover, data distribution normality was verified by visual inspection of histograms and QQ plots [6]. 

To assess sex and age group distribution among the participants in the half-marathon and marathon, we used a chi-square test (χ^2^). Specifically, we examined the association between participants’ sex and age group separately for each race, as well as between their sex and the race they participated in. The magnitude of these associations was tested by Cramer’s phi (φ), while the results were presented as a men-to-women ratio (MWR).

To test differences in PaceR between marathon and half-marathon runners in 9 age groups, 2 two-way analyses of variance (ANOVA) were performed (separately for men and women). Main effects of the race (marathon and half-marathon), age group (18–24; 25–29; 30–34; 35–39; 40–44; 45–49; 50–54; 55–59; 60+ years) and their interaction race x age group were performed. Additionally, 2 two-way analyses of variance (ANOVA) were performed (separately for men and women) to test differences in ES between marathon and half-marathon runners in 9 age groups. Main effects of race (marathon and half-marathon), age group (18–24; 25–29; 30–34; 35–39; 40–44; 45–49; 50–54; 55–59; 60+ years) and their interaction race x age group were performed. 

For all ANOVAs, the post-hoc Bonferroni test was performed. Effect size was presented via eta squared (ŋ^2^), where the values of 0.01, 0.06 and above 0.14 were considered small, medium, and large, respectively [27]. Alpha level was set at *p* < 0.05. All statistical tests were performed using Microsoft Office Excel 2007 (Microsoft Corporation, Redmond, WA, USA) and SPSS 20 (IBM, Armonk, NY, USA).

## 3. Results

The number of men and women in each race (i.e., men-to-women ratio – MWR) and their age group are presented in Table 1. As expected, more men participated in the 2017 Vienna Marathon than women (Total MWR was 2.44). A sex x race association was shown, (χ^2^ = 293.6, *p* < 0.01, φ = 0.13), where the men-to-women ratio was smaller in the half-marathon (1.99) in comparison to the marathon (3.77). Furthermore, sex x age group association was also shown in both the marathon (χ^2^ = 72.4, *p* < 0.01, φ = 0.11) and half-marathon (χ^2^ = 263.6, *p* < 0.01, φ = 0.15). In the marathon, the smallest MWR of 2.45 was observed in the 25–29 years age group, whereas the largest MWR of 8.35 was observed in the oldest age group. In the half-marathon, the smallest MWR of 1.14 was observed in the youngest age group, whereas the largest MWR of 3.96 was observed in the oldest age group.

The average running speeds for four sections and ES are presented in Table 2. From the descriptive data in Table 2, we can observe a gradual decrease in average speed throughout the race segments for both sexes in both the marathon and half-marathon, among all age groups. Furthermore, we can observe that ES are typically faster than Section 4, which has often been noted in marathon related studies. 

When PaceR in men runners was assessed (Figure 1), the results showed significant main effects of race [F_(17,12366)_ = 849.8, ŋ^2^ = 0.06, *p* < 0.01], age group [F_(17,12366)_ = 14.3, ŋ^2^ = 0.01, *p* < 0.01] as well as race x age group interaction [F_(17,12366)_ = 3.5, ŋ^2^ < 0.01, *p* < 0.01].

Bonferroni post-hoc test results for each race showed higher PaceR in marathon runners in comparison to half-marathon runners for all age groups (*p* < 0.01). Regarding age groups, all marathon runners younger than 30 years of age showed significantly higher PaceR than all runners from 30 to 59 years of age (*p* < 0.01). Moreover, marathon runners from the 60+ years group had higher PaceR than runners from 35 to 49 years of age (*p* < 0.01). Finally, half-marathon runners younger than 24 had higher PaceR than all runners from 30 to 54 years of age (*p* < 0.01).

When PaceR in women runners was assessed (Figure 2), the results showed significant main effects of race (F_(17,5063)_ = 79.3, ŋ^2^ = 0.02, *p* < 0.01) and age group (F_(17,5063)_ = 3.5, ŋ^2^ = 0.01, *p* = 0.01). A significant main effect of race x age group interaction was not obtained (F_(17,5063)_ = 0.62, ŋ^2^ < 0.01, *p* = 0.76).

Bonferroni post-hoc test results for each race showed higher PaceR in marathon runners in comparison to half-marathon runners for all age groups (*p* < 0.05). No significant differences were observed when the Bonferroni post-hoc test was performed for age group.

When the ES in men runners was assessed (Figure 3), the results showed significant main effects of race [F_(17,12366)_ = 25.5, ŋ^2^ < 0.01, *p* < 0.01] and age group [F_(17,12366)_ = 13.6, ŋ^2^ = 0.01, *p* < 0.01] whereas a significant main effect of race x age group interaction [F_(17,12366)_ = 0.97, ŋ^2^ < 0.01, *p* = 0.46] was not observed.

Bonferroni post-hoc test results for each race showed higher ES in half-marathon runners aged 25–49 years (*p* < 0.01). Moreover, half-marathon runners from 55–59 years of age showed higher ES than marathon runners of the same age (*p* < 0.05). Regarding Bonferroni post-hoc test results for age group, both marathon and half-marathon runners younger than 24 years showed significantly higher ES than all runners older than 30 years of age (*p* < 0.01). Moreover, half-marathon runners aged 24–29 years showed significantly higher ES than all runners older than 35 years of age (*p* < 0.01). Finally, half-marathon runners aged 30–34 years showed significantly higher ES than 45–49 year-old runners as well as the 60 + age group (all *p* < 0.01).

When ES in women runners was assessed (Figure 4), the results showed a significant main effect of age group (F_(17,5063)_ = 5.7, ŋ^2^ = 0.01, *p* < 0.01). Significant main effects of race (F_(17,5063)_ = 0.61, ŋ^2^ < 0.01, *p* = 0.43) and race x age group (F_(17,5063)_ = 1.7, ŋ^2^ < 0.01, *p* = 0.09) were not obtained.

Regarding Bonferroni post-hoc test results for age group, age groups <24 and 25–29 years showed significantly higher ES than all runners older than 35 (*p* < 0.01) in the half-marathon. Furthermore, half-marathon runners from 30–34 years showed significantly higher ES than all runners older than 45 (*p* < 0.01). Finally, runners older than 60 years showed significantly lower ES (*p* < 0.05) than runners from 35 to 49 years of age.

## 4. Discussion

The main aim of this study was to assess differences in pacing between marathon and half-marathon events in regards to the runners’ age groups, separately for men and women. In both men and women, regardless of age group, marathon runners showed greater variability in pacing than half-marathon runners. Furthermore, women showed no differences in pace variability with regard to the age group, whereas men younger than 30 years of age, as well as older men (over the age of 60), showed greater variability in pace than other age groups. Finally, younger half-marathon men and women showed the fastest end spurts compared to the older age groups and marathon runners.

Although more men participate in long distance events than women, several recent studies have reported a significant increase in women participants over the previous decades [28,29]. In this study, total MWR was 2.44. Moreover, the MWR was smaller in younger age groups, particularly in the half-marathon, where the lowest MWR was 1.14. The rationale for this can be found in fewer opportunities for older women to participate in sports [30], whereas a greater number of younger women have started to engage in regular training and sport in recent years [28]. Furthermore, the higher MWR in marathons might be explained by social discrimination and stricter medical caution for women than for men [31]. In addition, motivational factors, such as aspirations to have fun and stay healthy could justify the increase in women participating in half-marathons in comparison to men [32,33].

As seen in Table 2, positive pacing strategies (i.e., running speed decreased through the race) were observed in both marathon and half-marathon events, in all men and women age groups, which is in line with similar studies [19,22]. Moreover, positive pacing strategies often involve end spurts in the final sections of marathons and half-marathons [6,34], which was observed in this study as well. Positive pacing strategies with a quicker pace at the beginning segments of the race can often be explained as a result of “pre-race enthusiasm” or current absence of fatigue [4]. As a result, fatigue is more likely to occur later in race, thus reducing the running speed [8].

In all age groups, marathon runners (both men and women) showed significantly greater PaceR than half-marathoners. More even pacing in the half-marathon than the marathon indicates that aging plays a more important role as the race distance increases [19]. This phenomenon could exist due to the occurrence of severe fatigue in the marathon race after the 35th kilometer, which is often caused by glycogen depletion [11] rather than some psychological factors (e.g., fast race start due to competitiveness) [6]. However, the psychological characteristics of younger male runners (particularly younger than 30) could result in more variability in pacing. This variability appears primarily as a result of a fast start induced by their higher levels of self-esteem, which results in overestimation of their performance capabilities [26]. Moreover, younger, less experienced long distance runners could face problems with the control mechanism responsible for pacing, thus changing pace more often, which can result in rapid increase of fatigue. The important factors in setting an overall pacing strategy for a bout of exercise include knowledge of the endpoint and the associated duration of the event, an internal clock using scalar timing, and the memory of pacing strategy from prior events [3]. Younger runners with insufficient experience, could lack this “pacing template” in the brain, hence they change pace more often. On the other hand, elderly men spend more time running (i.e., run slowly). As a consequence, fatigue as well as increased pacing variability is more likely to occur.

Pacing in the women’s marathon and half-marathon does not seem to be influenced by age, which can be attributed to a better fatigue tolerance, regardless of age. Women generally have more fatigue-resistant type I muscle fibers, and are able to better utilize fat when participating in long distance running [14,35]. In addition, women are less likely to start a race at high-speed. The rationale for this can be found in women’s motivation for running long distance races. They are less competitive than men, and their motivation is to socialize, enjoy the race, and eventually, cross the finish line [33,36]. Nevertheless, the trend observed in Figure 2, showed that young and elderly women do have greater PaceR, particularly in marathons (however it was not statistically significant). The aforementioned findings could help sports and medicine practitioners to create separate age specific training plans and pacing strategies for men and women. This approach could help long distance runners to improve their physical fitness. Consequently, they could achieve better race times, reduce the potential risk of musculoskeletal injuries and increase the overall pleasure of long distance running.

Contrary to our previous study involving the Ljubljana half-marathon [19], an end spurt was observed in the Vienna marathon and half-marathon races. Race configuration, altitude, weather conditions, as well as the number of participants are possible explanations for these diametrically opposite outcomes. As a result, further studies of end spurts in marathons, and particularly half-marathons are required. In essence, the occurrence of a final spurt might be due to psychological factors. We have stated earlier, that pacing is a combination of anticipation, knowledge of the end-point, prior experience and an internal clock using scalar timing [3,37]. The knowledge of the near finish might motivate the runners to mobilize the last reserves. In particular, younger half-marathon men and women runners demonstrated the faster end spurts compared to the older age groups and marathon runners. These results further support our observation that younger long distance runners produce more variability in pacing. As a result, the end spurt was as much as 3.83% faster than Section 4 (in half-marathon women). Sports and medicine practitioners could use these findings to further familiarize young long distance runners with the most optimal pacing strategies, with the aim of enhancing performance, as well as reducing the risk of cardio-vascular emergencies, which are often induced by a fast end-spurt or high pace variability. Finally, race organizers and official race medical services could also benefit from these findings. Namely, they can spread awareness of the negative consequences of fast end spurts and high pace variability to race participants, thus preventing possible medical emergencies.

A limited number of participants in age groups older than 60 years prevented inclusion of older age groups and therefore limits our knowledge about pacing strategies in older runners. Furthermore, the absence of additional factors that could influence pacing, such as running experience, previous training routines, runners’ personal characteristics, running in large groups for social interaction, or limited weather information, can be considered as study limitations. Finally, this study has included only one event held in one year (i.e., the 2017 Vienna City Marathon), thus limiting the potential generalization of the findings. Conversely, the greatest strength of this study was its novelty, since it adds original information to the existing literature regarding one of the most popular race distances.

## 5. Conclusions

To conclude, in both men and women regardless of age group, marathon runners showed greater variability in pacing than half-marathon runners. In addition, women showed no differences in pace variability with regard to age group whereas the youngest and oldest men showed the greatest variability in pace. Finally, younger half-marathon men and women runners showed the fastest end spurts compared to older age groups and marathon runners. The presented findings could help sports and medicine practitioners to create separate age specific training plans and pacing strategies for men and women. This approach could help long distance runners to adopt an even or negative pacing profile with a low to moderate end spurt. This pacing approach could improve physical fitness, help runners to achieve better race times, reduce the potential risk of musculoskeletal injuries, and increase the overall pleasure in long distance running.

## Figures and Tables

**Figure 1 medicina-55-00479-f001:**
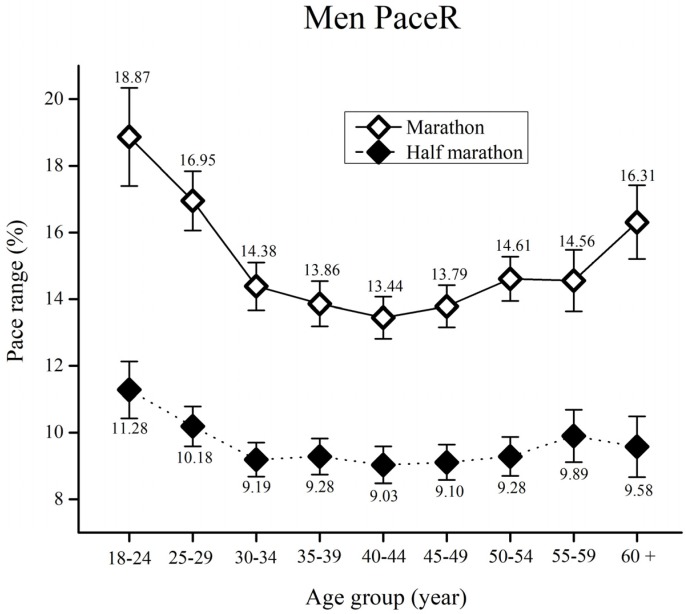
Men’s pace range (%) by age group for the marathon and the half-marathon. Error bars represent 95% confidence intervals.

**Figure 2 medicina-55-00479-f002:**
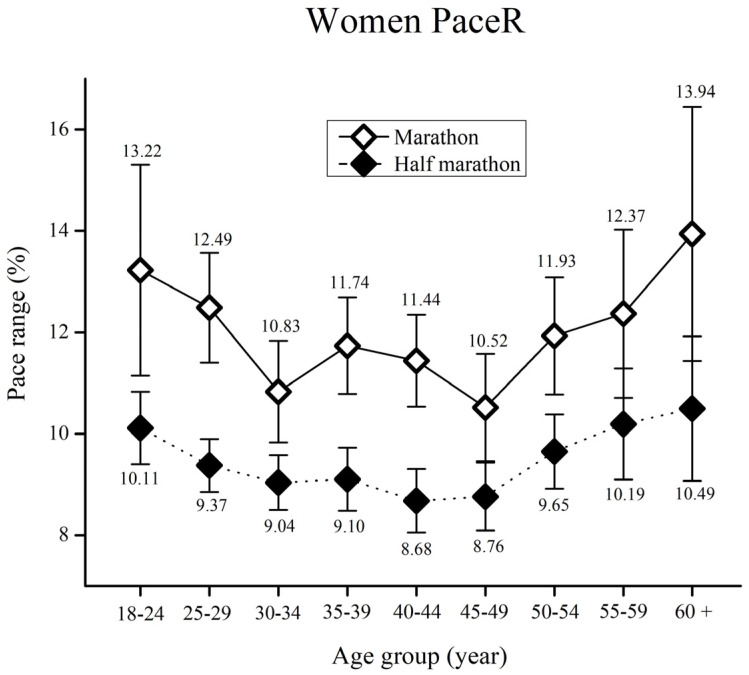
Women’s pace range (%) by age group for the marathon and half-marathon. Error bars represent 95% confidence intervals.

**Figure 3 medicina-55-00479-f003:**
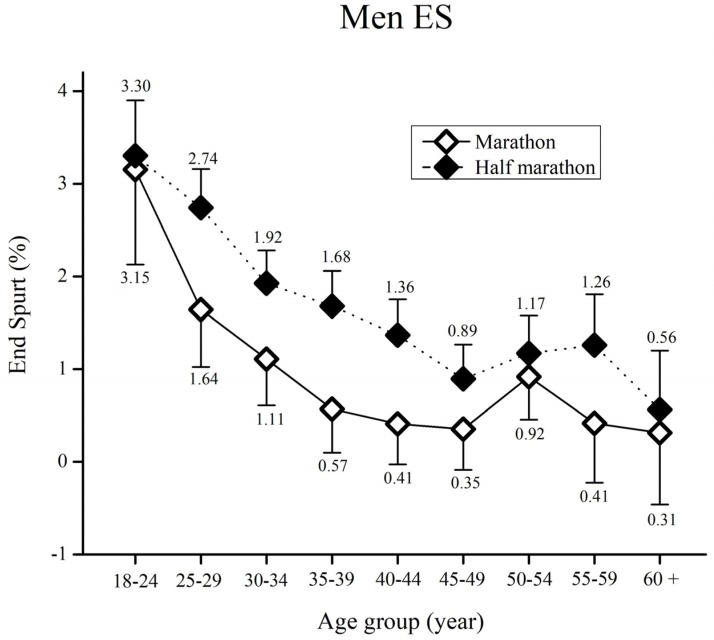
Men’s end spurt (%) by age group for the marathon and half-marathon. Error bars represent 95% confidence intervals.

**Figure 4 medicina-55-00479-f004:**
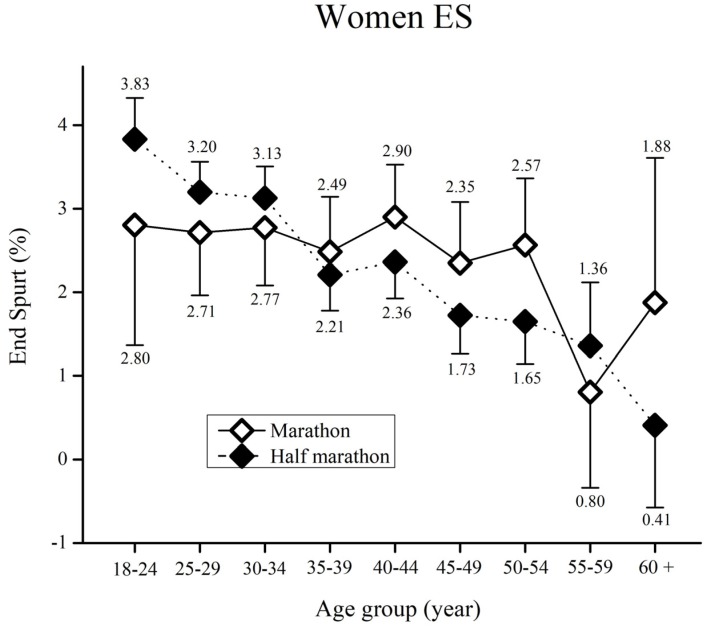
Women’s end spurt (%) by age group for the marathon and half-marathon. Error bars represent 95% confidence intervals.

**Table 1 medicina-55-00479-t001:** Number of participants and men-to-women ration in each race and age group.

Age Groups	Marathon				Half-Marathon			
	Men	Women	Total	MWR	Men	Women	Total	MWR
18–24	146	45	191	3.24	434	380	814	1.14
25–29	404	165	569	2.45	886	713	1599	1.24
30–34	618	194	812	3.19	1206	661	1867	1.82
35–39	694	214	908	3.24	1070	502	1572	2.13
40–44	798	237	1035	3.37	1032	496	1528	2.08
45–49	797	173	970	4.61	1128	433	1561	2.61
50–54	716	146	862	4.90	930	362	1292	2.57
55–59	373	71	444	5.25	513	162	675	3.17
60+	259	31	290	8.35	380	96	476	3.96
Total	4805	1276	6081	3.77	7579	3805	11,384	1.99

MWR = men-to-women ratio.

**Table 2 medicina-55-00479-t002:** Segments and race speed (m/s) for men and women, marathon and half marathon runners for each age group.

		Men				Women			
		Marathon		Half-Marathon		Marathon		Half-Marathon	
	Segment speed (m/s)	Mean	SD	Mean	SD	Mean	SD	Mean	SD
Age 18–24	Segment 1	3.18	0.50	3.20	0.49	2.94	0.41	2.87	0.35
	Segment 2	3.12	0.47	3.19	0.46	2.87	0.45	2.84	0.35
	Segment 3	3.05	0.51	3.17	0.48	2.88	0.51	2.80	0.37
	Segment 4	2.76	0.55	3.02	0.52	2.67	0.49	2.68	0.40
	End spurt	2.86	0.52	3.13	0.53	2.76	0.49	2.79	0.41
Age 25–29	Segment 1	3.28	0.56	3.20	0.48	3.00	0.41	2.88	0.33
	Segment 2	3.23	0.56	3.21	0.46	2.93	0.42	2.87	0.33
	Segment 3	3.18	0.61	3.19	0.48	2.91	0.44	2.84	0.36
	Segment 4	2.90	0.65	3.04	0.52	2.71	0.48	2.71	0.38
	End spurt	2.94	0.57	3.13	0.51	2.79	0.43	2.80	0.38
Age 30–34	Segment 1	3.27	0.52	3.22	0.47	2.99	0.40	2.89	0.31
	Segment 2	3.22	0.52	3.22	0.45	2.93	0.41	2.87	0.32
	Segment 3	3.19	0.54	3.21	0.47	2.91	0.43	2.85	0.34
	Segment 4	2.92	0.58	3.07	0.51	2.75	0.44	2.72	0.38
	End spurt	2.96	0.53	3.13	0.51	2.83	0.40	2.81	0.37
Age 35–39	Segment 1	3.30	0.47	3.18	0.46	2.99	0.43	2.89	0.36
	Segment 2	3.25	0.47	3.17	0.45	2.92	0.44	2.88	0.36
	Segment 3	3.22	0.49	3.15	0.46	2.90	0.46	2.85	0.39
	Segment 4	2.95	0.54	3.01	0.50	2.71	0.47	2.72	0.41
	End spurt	2.97	0.50	3.06	0.51	2.78	0.42	2.78	0.39
Age 40–44	Segment 1	3.26	0.42	3.17	0.44	2.88	0.29	2.86	0.33
	Segment 2	3.20	0.43	3.15	0.43	2.81	0.30	2.84	0.34
	Segment 3	3.18	0.45	3.14	0.44	2.79	0.33	2.82	0.35
	Segment 4	2.92	0.50	2.99	0.48	2.63	0.36	2.69	0.37
	End spurt	2.94	0.47	3.04	0.48	2.71	0.35	2.76	0.37
Age 45–49	Segment 1	3.21	0.40	3.15	0.42	2.88	0.31	2.84	0.29
	Segment 2	3.15	0.40	3.13	0.41	2.81	0.32	2.82	0.30
	Segment 3	3.12	0.43	3.11	0.43	2.80	0.35	2.79	0.32
	Segment 4	2.87	0.49	2.96	0.46	2.65	0.37	2.66	0.34
	End spurt	2.88	0.46	2.99	0.46	2.72	0.35	2.71	0.34
Age 50–54	Segment 1	3.13	0.39	3.10	0.41	2.85	0.28	2.81	0.28
	Segment 2	3.06	0.39	3.07	0.40	2.77	0.28	2.78	0.29
	Segment 3	3.02	0.42	3.04	0.43	2.75	0.31	2.74	0.31
	Segment 4	2.77	0.47	2.90	0.47	2.58	0.34	2.60	0.34
	End spurt	2.80	0.44	2.93	0.46	2.65	0.31	2.65	0.34
Age 55–59	Segment 1	3.10	0.35	3.01	0.39	2.84	0.27	2.77	0.26
	Segment 2	3.02	0.35	2.99	0.38	2.76	0.28	2.72	0.27
	Segment 3	2.99	0.37	2.95	0.40	2.75	0.29	2.68	0.28
	Segment 4	2.74	0.40	2.80	0.44	2.55	0.33	2.54	0.32
	End spurt	2.76	0.40	2.84	0.44	2.57	0.32	2.58	0.31
Age 60+	Segment 1	2.95	0.34	2.94	0.38	2.73	0.29	2.66	0.22
	Segment 2	2.87	0.36	2.91	0.39	2.63	0.31	2.62	0.25
	Segment 3	2.81	0.40	2.88	0.41	2.62	0.33	2.58	0.28
	Segment 4	2.57	0.43	2.74	0.43	2.41	0.36	2.45	0.29
	End spurt	2.58	0.41	2.75	0.43	2.45	0.34	2.46	0.30

SD = standard deviation.

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
