# Peer review of "Age Differences in Pacing in Endurance Running: Comparison between Marathon and Half-Marathon Men and Women"

_medicina, 2019, doi:10.3390/medicina55080479_

Round 1
Reviewer 1 Report
Very interesting manuscript and very well-written. The biggest concern that I have is that the title and the introduction section do not match the rest of the paper! It seems to be primarily focused on exploring sex differences, but the title/intro really just address the age group differences. I highly suggest adding some information on sex differences to the intro, and changing the title to better reflect the findings.
I have provided some specific comments below.
Abstract:
Line 18 – Where does the rationale for exploring sex differences come from? Seems out of place.
Line 21 – What is a “positive” pacing strategy?
I suggest making the Conclusions section shorter and increasing the details in the rest of the abstract.
Introduction:
Line 50 – What is meant by “the best”. I’m assuming for performance but it’s unclear.
Again, where is the rationale for exploring sex differences?
Methods:
Line 108 – I suggest making the Sections into bullet points, or placing them in a table.
Results:
Table 2 – I don’t think that many significant figures is really necessary. Going to just the tenths or hundredths place would clean up the table and make it easier to read.
Can the men’s and women’s pace range tables be placed side-by-side?
Author Response
Comments to the Author
Very interesting manuscript and very well-written. The biggest concern that I have is that the title and the introduction section do not match the rest of the paper! It seems to be primarily focused on exploring sex differences, but the title/intro really just address the age group differences. I highly suggest adding some information on sex differences to the intro, and changing the title to better reflect the findings.
Answer: Primarily, this manuscript focuses on exploring age differences as well as differences between marathon and half-marathon runners, independently for men and women. Therefore, we initially skipped mentioning men and women differences in introduction and title. However, obtained results led us to mention some of those differences in discussion and concussions. We are very thankful for these concerning comments from the expert reviewer and we changed the title and added some information on sex differences in the introduction. Finally, please note that the main aim of this study was to assess age and race (half-marathon vs. marathon) differences, and, therefore, we added rather limited information on sex differences.
Line 18 – Where does the rationale for exploring sex differences come from? Seems out of place.
Answer: We are thankful for this suggestion. Like we mention in the previous comment, the aim of this paper was not to assess sex differences. However, since it was previously showed that men and women differ in pacing, we did not want to assess age and race differences for men and women altogether, but separately. To better point that out, we changed abstract accordingly. Furthermore, we added some information in introduction as well.
Line 21 – What is a “positive” pacing strategy?
I suggest making the Conclusions section shorter and increasing the details in the rest of the abstract.
Answer: We are thankful for this suggestion and we changed it accordingly.
Line 50 – What is meant by “the best”. I’m assuming for performance but it’s unclear.
Again, where is the rationale for exploring sex differences?
Answer: We agree with the expert reviewer and we changed it accordingly. Moreover, in accordance with reviewers’ first comment, we added some rationale (in introduction) why we assessed age and race differences separately for men and women.
Line 108 – I suggest making the Sections into bullet points, or placing them in a table. Answer: We are thankful for this suggestion and we changed it according to the expert reviewer.
Table 2 – I don’t think that many significant figures is really necessary. Going to just the tenths or hundredths place would clean up the table and make it easier to read.
Answer: We are very thankful for this proposal and we changed it accordingly.
Can the men’s and women’s pace range tables be placed side-by-side?
Answer: We are very thankful for this proposal. Since we have investigated age and race pacing independently for men and women, our concept of the paper was to separate this figures. However, if the expert reviewer thinks this will considerable benefit the manuscripts, we are more than willing to do so.
Reviewer 2 Report
Based on the ongoing phenomena of increasing numbers of paticipants in endurance events the study covers an area of interest for athletes, coaches, race organizers and race physicians.
The introduction, methods and the results are well described / presented. The discussion and conclusions reflect well the results.
Major points, which should be analysed in relation to pacing strategies and endspurts, are the "time zones" and "time borders" of the races. Please therefore include - if possible with these datasets - an ANOVA for the main effect of "time zones" (finish times < 3 h, 3- 4 h, 5- 6 h and > 6 h for the marathon and < 1.5 h, 1.5 - 2 h, > 2 h for the half-marathon, respectively). If the numbers of participants are sufficient enough, the pacing and endspurts strategies for participants breaking "time barriers" like 3 h, 4, 5 and 6 hours in the marathon race and 1.5, 2 and 3 hours in the half-marathon, respectively, should be compared.
There are only few further suggestions:
Please mention the grade of humidity in relative and in absolute values, because air at that low temperatures (7.8 to 11.8° C) contains often a lower absolute amount of water than air at higher temperature, but the relative humidity rises with the decline of air temperature.
Please name the unit of speed in table 2 (m/s).
Table 2: two digits after the decimal point seem to be accurate enough.
Please include in your discussion / conclusion, that an endspurt and a higher pacing variability could also have an impact on the incidence of cardio-vascular emergencies.
Another hint: the knowledge of such variables like endspurt and a higher pacing variability should be incorporated in the plans of medical services for such races.
Please mention, that pacing strategies at international endurance events like the Vienna Marathon are also due to social interactions (e.g. bias of participating travelling groups, etc.).
Author Response
Based on the ongoing phenomena of increasing numbers of paticipants in endurance events the study covers an area of interest for athletes, coaches, race organizers and race physicians.
The introduction, methods and the results are well described / presented. The discussion and conclusions reflect well the results.
Answer: We are very thankful for these kind comments from the expert reviewer.
Major points, which should be analysed in relation to pacing strategies and endspurts, are the "time zones" and "time borders" of the races. Please therefore include - if possible with these datasets - an ANOVA for the main effect of "time zones" (finish times < 3 h, 3- 4 h, 5- 6 h and > 6 h for the marathon and < 1.5 h, 1.5 - 2 h, > 2 h for the half-marathon, respectively). If the numbers of participants are sufficient enough, the pacing and endspurts strategies for participants breaking "time barriers" like 3 h, 4, 5 and 6 hours in the marathon race and 1.5, 2 and 3 hours in the half-marathon, respectively, should be compared.
Answer: We are thankful for this note from the expert reviewer. We do have these data sets; however, adding such analysis involves further changes in title and abstract, lengthening the introduction and discussion to support time zones topic, as well as adding new figure, tables and references to this manuscript. Our humble opinion is that proposed analysis complies with major revision and not minor revision of this paper, as the journal editor suggested. Furthermore, adding all of this data would considerably lengthen the entire manuscript, since we planned to use this data for the entire new manuscript. Finally, if the expert reviewer thinks this analysis will noticeably benefit this manuscript, we are more than willing to do major revision.
Please mention the grade of humidity in relative and in absolute values, because air at that low temperatures (7.8 to 11.8° C) contains often a lower absolute amount of water than air at higher temperature, but the relative humidity rises with the decline of air temperature.
Answer: We are thankful for this suggestion, however, no additional information on humidity grade or wind speed was available on the official race web address, only what we stated in the manuscript. We pointed that out in the method section, and listed that as one of the study limitations.
Please name the unit of speed in table 2 (m/s).
Answer: We are thankful for this suggestion and we changed it according to the expert reviewer.
Table 2: two digits after the decimal point seem to be accurate enough.
Answer: We are thankful for this suggestion and we changed it accordingly.
Please include in your discussion / conclusion, that an endspurt and a higher pacing variability could also have an impact on the incidence of cardio-vascular emergencies.
Answer: We are very thankful for this proposal and we added this to the discussion section.
Another hint: the knowledge of such variables like endspurt and a higher pacing variability should be incorporated in the plans of medical services for such races.
Answer: We fully agree with the expert reviewer and we added this to the discussion section.
Please mention, that pacing strategies at international endurance events like the Vienna Marathon are also due to social interactions (e.g. bias of participating travelling groups, etc.).
Answer: We fully agree with the expert reviewer and we added this to the limitation section
Reviewer 3 Report
Authors examined the age differences in pacing in endurance running analyzing the race performance of men and women participating in the Vienna City marathon and half-marathon. Overall the paper is well organized and well written. The methods are adequate, tables and figures clearly indicate the findings, the discussion is properly related to the results and the conceptual framework.
Author Response
Comments to the Author
Authors examined the age differences in pacing in endurance running analyzing the race performance of men and women participating in the Vienna City marathon and half-marathon. Overall the paper is well organized and well written. The methods are adequate, tables and figures clearly indicate the findings, the discussion is properly related to the results and the conceptual framework.
Answer: We are very thankful for these kind comments from the expert reviewer.